# Multiparametric Magnetic Resonance Imaging Correlates of Isocitrate Dehydrogenase Mutation in WHO high-Grade Astrocytomas

**DOI:** 10.3390/jpm13010072

**Published:** 2022-12-29

**Authors:** Arpita Sahu, Nandakumar G. Patnam, Jayant Sastri Goda, Sridhar Epari, Ayushi Sahay, Ronny Mathew, Amit Kumar Choudhari, Subhash M. Desai, Archya Dasgupta, Abhishek Chatterjee, Pallavi Pratishad, Prakash Shetty, Ali Asgar Moiyadi, Tejpal Gupta

**Affiliations:** 1Neuro-Oncology Disease Management Group, Tata Memorial Centre, Mumbai 400012, India; 2Department of Radiodiagnosis, Tata Memorial Centre, Mumbai 400012, India; 3Homi Bhabha National Institute, Mumbai 400012, India; 4Department of Radiation Oncology, Tata Memorial Centre, Mumbai 400012, India; 5Department of Pathology, Tata Memorial Centre, Mumbai 400012, India; 6Department of Biostatistics, Tata Memorial Centre, Mumbai 400012, India; 7Department of Neurosurgery, Tata Memorial Centre, Mumbai 400012, India

**Keywords:** IDH (Isocitrate dehydrogenase)-mutants, necrosis, rCBV (relative cerebral blood volume), gliomas, grade-4, Glioblastomas

## Abstract

Purpose and background: Isocitrate dehydrogenase (IDH) mutation and O-6 methyl guanine methyl transferase (MGMT) methylation are surrogate biomarkers of improved survival in gliomas. This study aims at studying the ability of semantic magnetic resonance imaging (MRI) features to predict the IDH mutation status confirmed by the gold standard molecular tests. Methods: The MRI of 148 patients were reviewed for various imaging parameters based on the Visually AcceSAble Rembrandt Images (VASARI) study. Their IDH status was determined using immunohistochemistry (IHC). Fisher’s exact or chi-square tests for univariate and logistic regression for multivariate analysis were used. Results: Parameters such as mild and patchy enhancement, minimal edema, necrosis < 25%, presence of cysts, and less rCBV (relative cerebral blood volume) correlated with IDH mutation. The median age of IDH-mutant and IDH-wild patients were 34 years (IQR: 29–43) and 52 years (IQR: 45–59), respectively. Mild to moderate enhancement was observed in 15/19 IDH-mutant patients (79%), while 99/129 IDH-wildtype (77%) had severe enhancement (*p*-value <0.001). The volume of edema with respect to tumor volume distinguished IDH-mutants from wild phenotypes (peritumoral edema volume < tumor volume was associated with higher IDH-mutant phenotypes; *p*-value < 0.025). IDH-mutant patients had a median rCBV value of 1.8 (IQR: 1.4–2.0), while for IDH-wild phenotypes, it was 2.6 (IQR: 1.9–3.5) {*p*-value = 0.001}. On multivariate analysis, a cut-off of 25% necrosis was able to differentiate IDH-mutant from IDH-wildtype (*p*-value < 0.001), and a cut-off rCBV of 2.0 could differentiate IDH-mutant from IDH-wild phenotypes (*p*-value < 0.007). Conclusion: Semantic imaging features could reliably predict the IDH mutation status in high-grade gliomas. Presurgical prediction of IDH mutation status could help the treating oncologist to tailor the adjuvant therapy or use novel IDH inhibitors.

## 1. Introduction

The management of high-grade gliomas has undergone a paradigm shift with the addition of molecular parameters to morphological features in the World Health Organization (WHO) 2016 classification [1]. The modern-day treatment practices of gliomas are now based on molecular biomarkers to have a biologically homogenous treatment group to study newer interventions in clinical trials. Although previously categorized under grade 4 gliomas as an isocitrate dehydrogenase (IDH) mutant and wildtype, glioblastomas are now considered biologically and molecularly separate entities: glioblastoma IDH-wildtype and IDH-mutant grade 4 astrocytoma [2]. Molecular biomarkers such as IDH and MGMT (O-6 methylguanine methyltransferase) have allowed oncologists to personalize treatment and prognosticate the disease better than it used to be a decade earlier [3].

Despite multimodal therapy that includes gross total resection (GTR), radiotherapy, and chemotherapeutics such as temozolomide (TMZ), the prognosis for grade 4 gliomas is dismal with a median survival of only 14 months, and <5% of patients survive beyond five years [1]. This calls for a more concerted approach to understanding the disease biology and the factors associated with its outcome. Literature studies have shown that age, performance status, and treatment-related factors can prognosticate these tumors. However, these factors lack the accuracy to predict response to therapy [4]. With a better understanding of biology and the advent of newer molecular techniques, researchers have been able to show unique biomarkers that could predict treatment response and prognosticate these tumors with a high degree of accuracy, paving the way for a more personalized treatment approach. The two molecular biomarkers of significant interest that have translated into clinical practice are IDH and MGMT, which are responsible for the epigenetic alterations in grade 4 gliomas. Evaluating these biomarkers have become a norm in tailoring therapy and disease prognostication [5].

IDH plays an essential role in the Kreb’s cycle by converting isocitrate to alpha-ketoglutarate. Mutated IDH converts alpha-ketoglutarate into 2-hydroxyglutarate, an oncometabolite responsible for the epigenetic changes in gliomas and associated with improved prognosis. Therefore, IDH-mutants have a better prognosis compared to wild phenotypes. Thus, IDH is a diagnostic and prognostic biomarker for gliomas [6]. Mutated IDH also drives increased methylation in gliomas [7]. MGMT is a DNA (deoxyribonucleic acid) repair enzyme, detoxifying temozolomide (TMZ)-induced DNA damage. Clinically, high MGMT protein expression has been associated with therapeutic resistance to DNA-alkylating agents apart from having prognostic significance [8].

Typically, these markers can only be assessed from the tumor tissues of the surgical specimen. However, it is well-known that grade 4 gliomas are genetically heterogenous, and targeting may lead to erroneous interpretations [9]. Therefore, using high-quality non-invasive magnetic resonance imaging (MRI) parameters to distinguish grade 4 gliomas with different genetic compositions (radiogenomics) has gained importance for prognostication and therapy selection [10]. Imaging studies have used VASARI (Visually AcceSAble Rembrandt Images) semantic features for the molecular subgrouping of grade 4 gliomas [11].

Given the inherent tumor heterogeneity on histopathology and the universal availability of MRI, we envisaged multiparametric semantic MRI features in a large cohort of grade 4 glioma patients to classify them based on the IDH status as confirmed by gold standard immunohistochemistry (IHC) with or without gene sequencing.

## 2. Materials and Methods

### 2.1. Patient Selection

This was a retrospective chart review of 148 grade 4 glioma patients in whom we had the baseline MRI and IDH mutational status. The study was carried out after approval from the Institutional Ethics Committee (IEC). One hundred and forty-eight patients were analyzed after assessing the eligibility criteria: availability of baseline pre-treatment MRI and IDH mutational status in our hospital’s electronic medical records (EMR).

### 2.2. MR Protocol

The MRI sequences performed were: T2 fast spin-echo (time to repeat/time to echo–TR/TE: 2700/100), fluid attenuation and inversion recovery–FLAIR (TR/TE: 9000/120; inversion time: 2200 ms), unenhanced T1 spin-echo and three planes of contrast-enhanced T1 spin-echo (TR range/TE: 600–700/20), axial gradient-echo images (TR/TE: 570/30), and axial diffusion-weighted images (TR/TE: 8300/70; b-1000 s/mm^2^; 4-mm thickness). Dynamic susceptibility contrast-enhanced (DSC) perfusion imaging was performed with the first-pass acquisition of gadolinium-based contrast injected at 0.2 mmol/kg, followed by a saline chaser using a power injector. The processing of perfusion images for relative cerebral blood volume (rCBV) was performed using a leakage correction algorithm.

### 2.3. MR Imaging Parameters

We adopted the standard VASARI imaging features with an expanded set of parameters to study their association with IDH status [12]. The VASARI features and extended imaging parameters are provided in Appendix A.

Pre-treatment multiparametric MRI was assessed for tumor location, laterality (right, left, or midline), enhancement pattern subcategories I, II, and III (I–mild/moderate and severe: based on enhancement intensity, II–homogeneous/heterogeneous enhancement, and III–rim/nodular/solid/patchy type of enhancement), proportion of enhancing tumor (<25%, 25–50%, and >50% of tumor), eloquent cortex involvement, percentage necrosis (nil, <25%, 25–50%, >50% of tumor volume), proportion of edema (<tumor volume, equal to tumor volume, >tumor volume), presence/absence of hemorrhage, presence/absence of cysts (if present, whether cyst was with hemorrhage), unifocal/multifocal/multicentric, tumor size (<2 cm, 2–5 cm, and >5 cm in its longest dimension), presence of satellite lesions, presence of leptomeningeal spread, midline shift (absent, mild: <5 mm, moderate: 5–10 mm, and severe >10 mm), tumor crossing midline, presence of calvarial remodelling, presence of restricted diffusion, presence of ependymal invasion, epicenter (cortical gray matter/deep white matter) with presence of subcortical involvement, tumor margins (well-defined, ill-defined, and well-defined with areas of focal infiltration), presence of FLAIR/T2 mismatch (homogeneous or heterogeneous signal intensity on T2WI and relative hypointensity on FLAIR with a peripheral hyperintense rim), rCBV value, and closeness to the subventricular zone (Appendix B). The radiologist reviewing the imaging features was blinded to the patient’s details and IDH status.

### 2.4. Mutational Analysis

Patients’ surgical pathology reports from electronic medical records (EMR) were used for IDH mutational status. IDH status was available for all 148 grade 4 glioma patients. The standard institutional practice is to perform immunohistochemistry for IDH status, which was considered the gold standard and confirmed by Sangers sequencing for IDH1R132 and IDH2R172 loci if required.

### 2.5. Statistical Analysis

Statistical analysis was performed using the Statistical Package for the Social Sciences software (SPSS, ver. 21). Descriptive statistics such as frequencies and percentages were used for the categorical variables, and mean and standard deviation (SD) for the continuous variables. Univariate analyses were done using the Fisher’s exact test or the chi-square test. Multivariate stepwise logistic regression analyses were conducted to evaluate the risk factors for IDH mutation status. Variables with a *p*-value of < 0.05 on the univariate analysis were used as inputs for the multivariable analysis. A *p*-value < 0.05 was considered statistically significant.

## 3. Results

### 3.1. Patient Demographic and Tumor Characteristics

From the neuro-oncology database, 148 patients were taken up for the study. Most grade 4 glioma patients were IDH-wildtype (*n* = 129), while IDH-mutant phenotype was observed in only 19 patients, and their representative images are provided in Figure 1 and Figure 2. The male-to-female ratio was 2:1 (males: 99 and females: 49). The median age of patients who harbored IDH mutations was significantly lower than the patients who were IDH-wildtype. Specifically, for the IDH-mutant group, median age was 34 years (interquartile range, IQR: 29.5–43), while for the IDH-wildtype group, median age was 52 years (IQR: 45–59). We observed that 13/19 (68.4%) IDH-mutant patients had left hemispheric predominance than 55/129 IDH-wildtype patients (42.6%). The frontal lobe was predominantly involved in IDH-mutants, while multilobar involvement was more common in IDH-wildtypes. The patient and tumor demographic profiles are summarized in Table 1.

### 3.2. MRI Parameters of IDH-Wildtype vs. Mutant Phenotype Tumors

Univariate analysis of the semantic MRI features of IDH-mutant vis-à-vis IDH-wildtype grade 4 gliomas have been summarized in Table 2.

**Pre-contrast MRI parameters** (T1, T2, FLAIR, diffusion, and gradient echo): the absence of perilesional edema and edema volume less than the tumor volume (15.8% and 57.9%, respectively) were commonly associated with IDH-mutants. In comparison, IDH-wildtype tumors had perilesional edema volume equal to and more than tumor volume in 27.1% and 17.8% of patients, respectively (Figure 3). The overall presence of cysts (Figure 4) was observed in 22 patients (14.8%). It was predominant in IDH-mutant phenotypes (31.6%). However, hemorrhagic cysts (Figure 5) were observed in two patients of IDH-mutant phenotypes, which were conspicuously absent in IDH-wildtypes. Subcortical involvement (Figure 6) was predominant in IDH-mutants (94.7%).

Presence of T2-FLAIR mismatch (Figure 7) was seen in 15.8% of IDH-mutants and 3.9% of IDH-wildtypes (*p*-value 0.067), although not statistically significant. Other parameters such as tumor location, involvement of eloquent cortex, presence of hemorrhage, tumor size, tumor crossing midline, diffusion restriction, multicentricity/multifocality, and distance from subventricular zone did not correlate with IDH status.

**Post-contrast MRI T1 parameters**: among the IDH-mutants, nine (47.4%) had mild enhancement, six (31.6%) had moderate enhancement, and only four patients had severe enhancement. A proportion of 99/129 IDH-wildtypes (76.7%) were associated with severe enhancement (Figure 8). A patchy enhancement pattern was seen in 11/19 (57.9%) of patients with the IDH-mutant phenotype. However, rim enhancement was common in IDH-wildtypes (104/129; 80.6%) (Figure 9). In our cohort, absence of necrosis (15.8%) and < 25% necrosis (57.9%) was commonly associated with IDH-mutants (Figure 10). In comparison, IDH-wildtypes had predominantly 25–50% necrosis (27.1%) and >50% necrosis (59.7%). The presence of dural enhancement was found only in IDH-wildtypes (24%) (Figure 11).

Other parameters such as the proportion of the enhancing tumor, leptomeningeal spread, satellite lesions, calvarial remodeling, and ependymal invasion did not correlate with the IDH status. These are summarized in Appendix A.

**MRI perfusion parameter**: the rCBV values were available for 124/148 patients. IDH-mutants had a median rCBV of 1.8 (IQR: 1.4–2.0), which was significantly lower than the IDH-wildtypes with a median value of 2.6 (IQR: 1.9–3.5) (*p*-value 0.001).

On multivariate analysis, using a stepwise logistic regression, the MRI parameters of necrosis of > 25% on CE-MRI (*p*-value < 0.001) and rCBV cut-off of > 2.0 (*p*-value 0.007), independently correlated for IDH-wildtype phenotype while the other semantic features that were significant on the univariate analysis lost their significance on the multivariate analysis (Table 3). Figure 12 shows the graphical abstract for this study.

## 4. Discussion

This study attempts to identify the semantic imaging features on multiparametric MRI that best define IDH mutation status against the gold standard immunohistochemistry to predict these mutations. To the best of our knowledge, this study includes one of the largest cohorts of grade 4 gliomas with a description of its imaging morphology at diagnosis and radiopathological correlation.

In our cohort, we found that rCBV and the proportion of necrosis accurately predicts the IDH mutation status on multivariate analysis. The median rCBV for IDH-mutants was 1.8 (IQR: 1.4–2.0), which was significantly lower than the IDH-wildtypes with a median of 2.6 (IQR: 1.9–3.5) on univariate analysis. On multivariate analysis, the rCBV cut-off of <2.0 was able to differentiate IDH-mutants from IDH-wildtype tumors. Our results corroborate with various other studies [13,14,15,16]. Additionally, studies have shown a higher nCBV (normalized CBV) in IDH-wildtypes [17]. The hemodynamic tissue signature segmentation model explains the reduced rCBV in mutant phenotypes [18]. Literature studies have attributed the lower rCBV in IDH-mutants to reduced angiogenesis in the tumor tissue while a positive association of IDH-wildtype tumors have been attributed with increased necrosis [19]. In our study, the percentage of necrosis in IDH-mutant cohorts was <25% of the entire tumor volume, while in IDH-wild phenotypes necrosis was >25%. These findings corroborate the results of several studies [13,14,20]. Park et al. observed that a cut-off necrosis of <33% of the tumor was commonly associated with IDH-mutants [20]. The increased necrosis and non-enhancing tumors in wildtype phenotypes have been attributed to augmented intratumoral hypoxia that results from the activation of the coagulation pathway and intravascular thrombosis leading to excessive tumor necrosis [13,14].

We observed that a mild to moderate and patchy enhancement was commonly associated with the IDH-mutant phenotype tumors. Similarly, IDH-wildtype tumors had severe or intense rim enhancement, a notable feature documented in other literature studies [14,15,16]. Molecular- and genetic-level research provides few valid reasons for these significant associations. It has been observed that IDH-mutant tumors have reduced expression of vascular endothelial growth factor and thereby, reduced vascular permeability leading to reduced enhancement in mutant subtype tumors compared to wildtype tumors [15]. Minimal to mild contrast enhancement in IDH-mutants was attributed to a lack of microvascular proliferation resulting in reduced neoangiogenesis, vascular permeability, and contrast enhancement [13]. Secondly, IDH-wildtypes are known to have increased expression of hypoxia-inducible factor-1α (HIF-1α), which is associated with an increased expression of proangiogenic factors such as vascular endothelial growth factor-A (VEGF-A) and platelet-derived growth factor-A (PDGF-A), responsible for neoangiogenesis leading to increased contrast enhancement [14]. Carrillo et al. showed an increased association of rim enhancement in MGMT unmethylated patients [15].

The absence of peritumoral edema or edema volume less than tumor volume was commonly associated with IDH-mutants, while peritumoral edema volume equal to or more than tumor volume was associated with IDH-wildtypes. Similar results were documented in studies by Lasocki et al. [11] and Patel et al. [16]. Lasocki’s group obtained a cut-off value of 33% to differentiate IDH-mutant from IDH-wildtype tumors. IDH-mutants were associated with the presence of cysts, which is in concordance with other studies [21].

In the present study, left lobe predominance was associated with IDH-mutant tumors, which concurs with the observations of a prior survey by Ellingson et al. [22]. Left hemispheric predominance is a common finding in MGMT-methylated tumors. In our cohort, all the IDH-mutant patients had MGMT methylation. Although literature studies have no proper explanation for this phenomenon, studies have demonstrated that MGMT-methylated tumors have a left hemispheric predominance [22] and most of the MGMT-methylated tumors are IDH-mutant subtypes. Further research needs to be conducted to understand the predisposition of IDH-mutated and MGMT-methylated tumors toward the left hemisphere. Subcortical involvement in our study was primarily seen in IDH-mutants (94.7%). A plausible explanation for this association could be the preponderance of deep white matter involvement by wildtype tumors. Dural enhancement was absent in all IDH-mutants and was present in 24% of IDH-wildtype tumors, probably due to increased vascularity and dural invasion leading to increased dural enhancement. These observations in our study are novel and yet to be documented in the literature.

The T2-FLAIR mismatch sign was predominant in IDH-mutants, characterized by homogeneous or heterogeneous signal intensity on T2WI and relative central hypointensity on FLAIR with a peripheral hyperintense rim [16]. However, in the present study, 15.8% of IDH-mutants and 3.9% of IDH-wildtypes had a peripheral hyperintense rim.

Being a retrospective study, we acknowledge that our results are limited by the inherent bias associated with all retrospective studies. Study designs with still larger sample sizes or prospective studies are needed in this domain to address the inequalities between IDH-mutant and wildtype cohorts. Although a single institutional study, the results are derived from a large cohort of patients who had uniform preoperative MRIs and underwent molecular studies with the internationally recommended gold standard test. We obtained a significant correlation between IDH mutational status and various imaging parameters that corroborate the literature studies. We documented a few new and unique features such as enhancement patterns–rim/nodular/patchy/solid, dural enhancement, and subcortical involvement, which have been highlighted in our study.

## 5. Conclusions

Semantic imaging features reliably predicted the IDH status in patients diagnosed with grade 4 gliomas. The correlations were more robust when an advanced technique such as perfusion was employed. Tumor necrosis of <25% and rCBV values of <2.0 stood out as independent imaging surrogates for IDH mutation. With the evolution in glioma therapeutics, the advent of newer strategies, and the ongoing trials for targeted therapies, we envisage a need for molecular predictions based on fast, non-invasive, and easy-to-adopt semantic radiological features shortly to select outpatients for targeted therapeutic interventions.

## Figures and Tables

**Figure 1 jpm-13-00072-f001:**
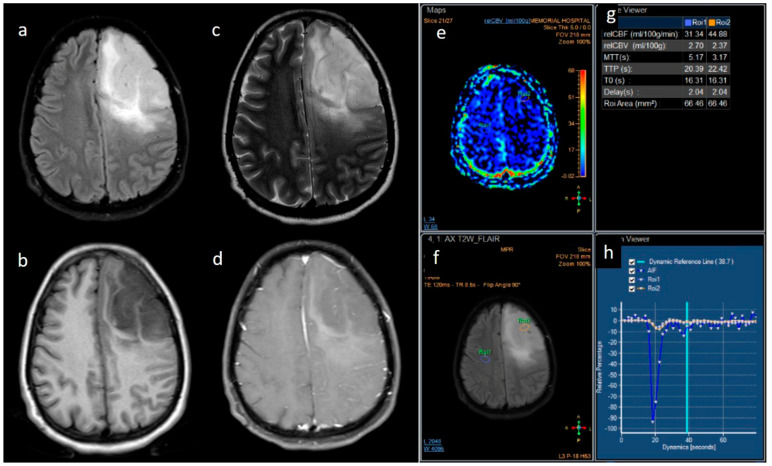
(**a**–**h**) FLAIR, T1, T2, post-contrast T1, perfusion color map, and graph representative of IDH-mutant grade 4 astrocytoma. Images a–d show a homogenous well-circumscribed mass without necrosis and edema less than the tumor volume. Images e–h are dynamic susceptibility contrast images showing hypoperfusion.

**Figure 2 jpm-13-00072-f002:**
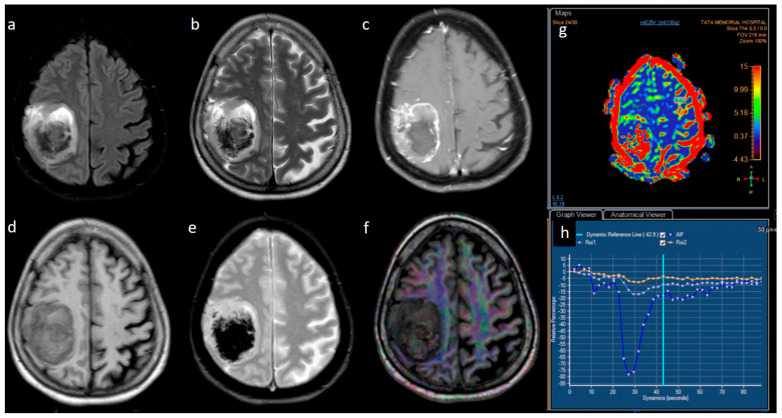
(**a**–**h**) FLAIR, T2, T1, post-contrast T1, GRE, DTI, perfusion color map, and graph representing an IDH-wildtype glioblastoma. Images a–e reveal a heterogeneous peripheral rim-enhancing right frontoparietal mass with significant internal necrosis and hemorrhage. Image f is a DTI image that shows subcortical and adjacent parenchymal infiltration. Images g and h reveal significant hyperperfusion.

**Figure 3 jpm-13-00072-f003:**
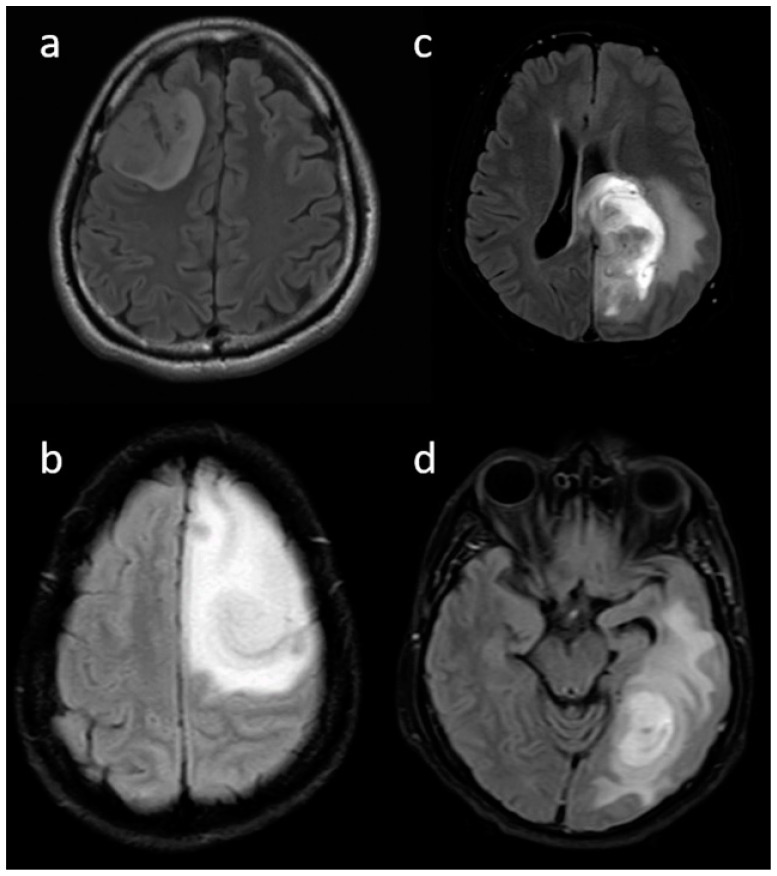
(**a**–**d**) Representative FLAIR images showing various proportions of edema: (**a**) none, (**b**) less than tumor volume, (**c**) equal to tumor volume, and (**d**) more than tumor volume.

**Figure 4 jpm-13-00072-f004:**
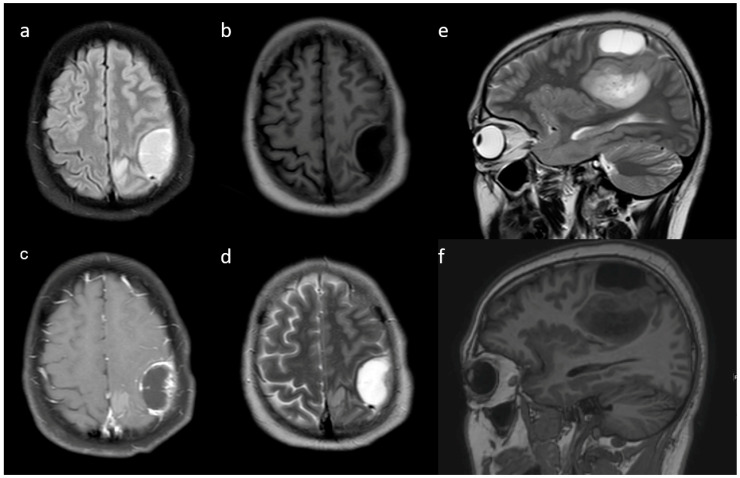
(**a**–**d**) Axial T1w, T2w, FLAIR, and post-contrast T1w images reveal a peritumoral cyst adjacent to the primary tumor. Additional (**e**,**f**) sagittal T2w and sagittal T1w are also shown.

**Figure 5 jpm-13-00072-f005:**
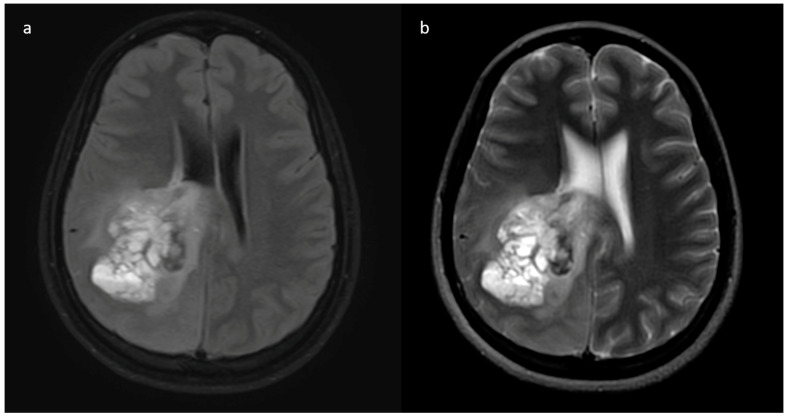
(**a**,**b**) Axial T2w and FLAIR images reveal intratumoral cysts with fluid-fluid levels within these cysts, suggestive of a hemorrhage.

**Figure 6 jpm-13-00072-f006:**
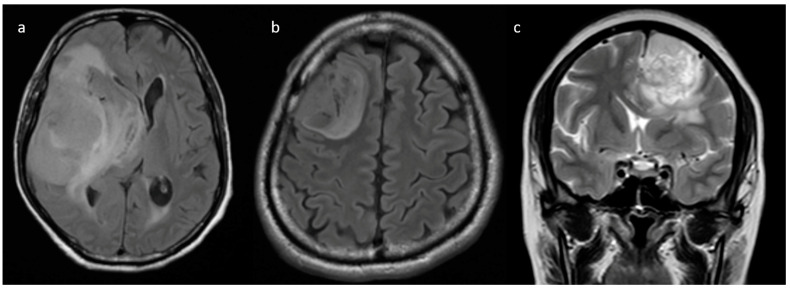
(**a**–**c**) Representative axial FLAIR and coronal T2w images show a tumor with a broad base towards the cortex with subcortical involvement.

**Figure 7 jpm-13-00072-f007:**
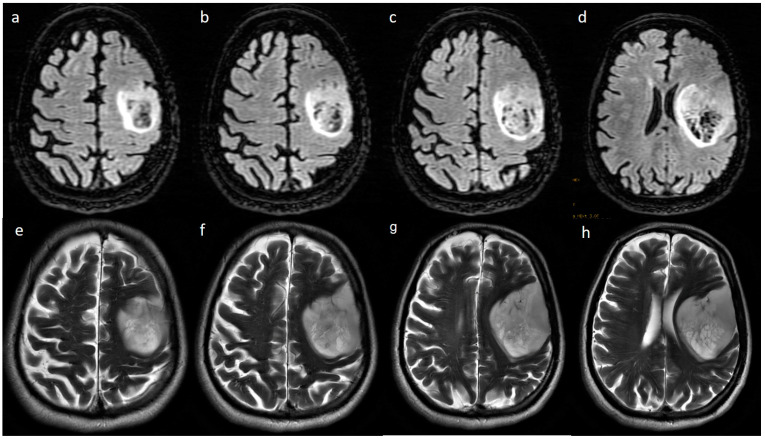
Axial FLAIR (**a**–**d**) and T2w (**e**–**h**) images reveal a T2-FLAIR mismatch sign in the form of hypointensity within the central part with the hyperintense rim on FLAIR, while both are hyperintense on T2.

**Figure 8 jpm-13-00072-f008:**
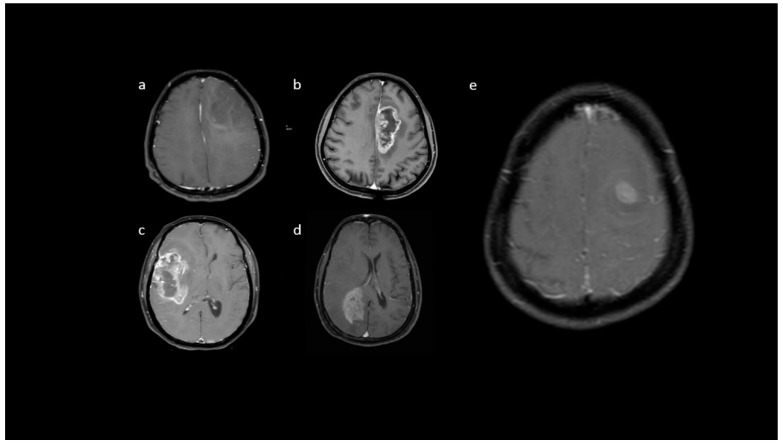
(**a**–**e**) Representative post-contrast T1 images showing various enhancement patterns: (**a**) no noticeable enhancement; (**b**,**c**) severe enhancement, (**d**) moderate enhancement pattern, and (**e**) mild enhancement.

**Figure 9 jpm-13-00072-f009:**
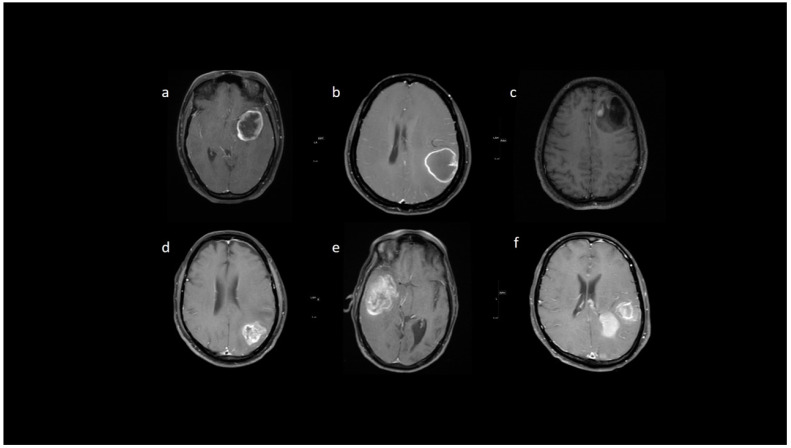
(**a**–**f**) T1 + C images show: (**a**,**b**) rim, (**c**) nodular, (**d**,**e**) patchy, and (**f**) solid enhancement pattern.

**Figure 10 jpm-13-00072-f010:**
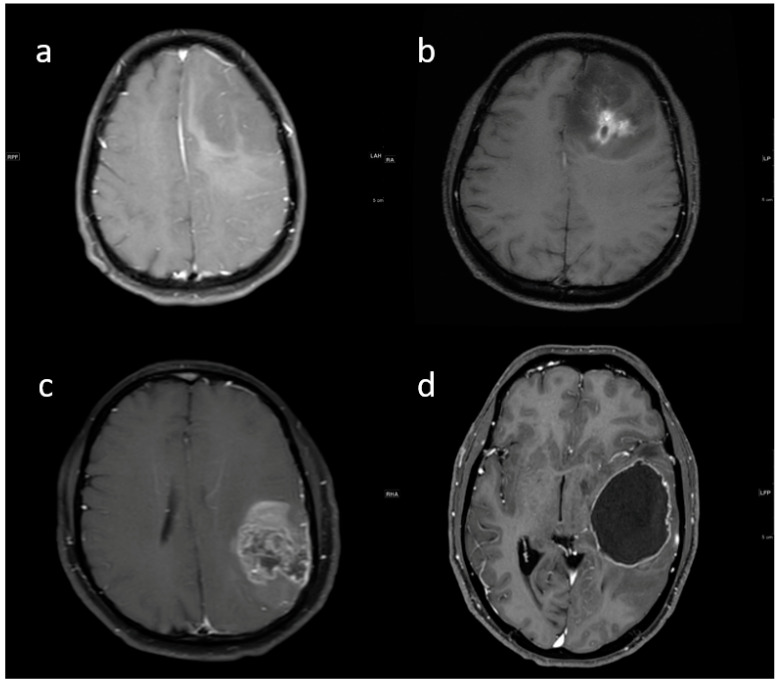
(**a**–**d**) T1 + C images from four patients showing various proportions of necrosis: (**a**) none, (**b**) < 25%, (**c**) 25–50%, and (**d**) > 50%.

**Figure 11 jpm-13-00072-f011:**
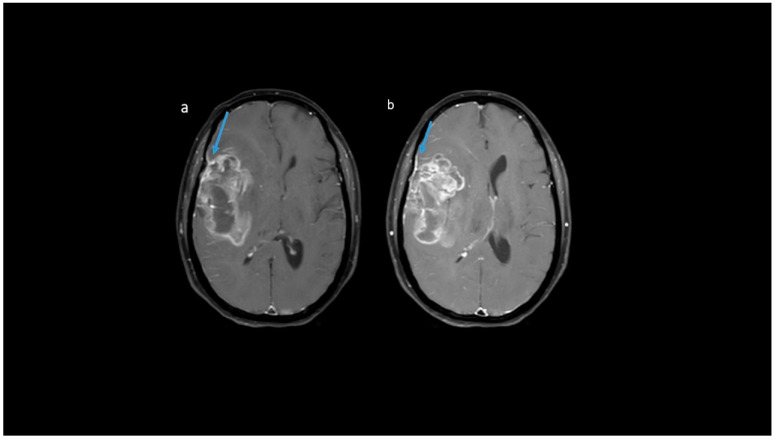
(**a**,**b**) Representative MR axial T1 + C images show dural enhancement along the right frontoparietal convexity in IDH-wildtype tumor.

**Figure 12 jpm-13-00072-f012:**
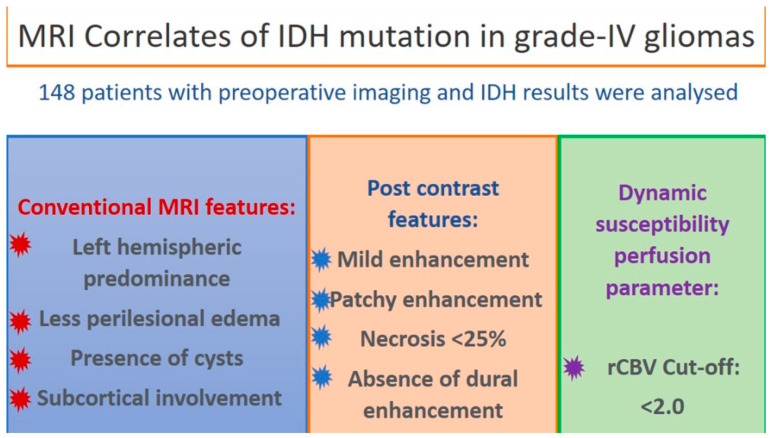
Pictorial depiction of the results of our study.

**Table 1 jpm-13-00072-t001:** Patient demographic and tumor profiles, and their correlation with IDH status.

Parameter	IDH-Mutant (N-19) Frequency (%)	IDH-Wildtype (N-129) Frequency (%)	*p*-Value
**Age (years)** Median (IQR)	34 (29.5–43)	52 (45–59)	<0.001
**Gender** Female Male	5 (26.3) 14 (73.7)	44 (34.1) 85 (65.9)	0.680
**Tumor location** Frontal Temporal Insular Occipital Parietal Brainstem/cerebellum Multiple sites	10 (52.6) 3 (15.8) 0 0 1 (5.3) 0 5 (26.3)	36 (27.9) 19 (14.7) 2 (1.6) 4 (3.1) 19 (14.7) 2 (1.6) 47 (36.4)	0.420
**Tumor laterality** Right Left Bilateral/central	3 (15.8) 13 (68.4) 3 (15.8)	64 (49.6) 55 (42.6) 10 (7.8)	0.020
**Tumor size** <5 cm >5 cm	4 (21.1) 15 (78.9)	51 (39.5) 78 (60.5)	0.136

IQR: interquartile range.

**Table 2 jpm-13-00072-t002:** Univariate analysis of MRI imaging correlates of IDH status.

Variable	Parameter	IDH-Mutated (*n* = 19)	IDH-Wildtype (*n* = 129)	*p*-Value
Enhancement I	Mild	9 (47.4)	5 (3.9)	<0.001
	Moderate	6 (31.6)	25 (19.4)	
	Severe	4 (21.1)	99 (76.7)	
Enhancement III	Rim	6 (31.6)	104 (80.6)	<0.001
	Nodular	0	2 (1.6)	
	Patchy	11 (57.9)	13 (10.1)	
	Solid	2 (10.5)	10 (7.8)	
Necrosis	None	3 (15.8)	3 (2.3)	<0.001
	<25%	11 (57.9)	14 (10.9)	
	25- 50%	2 (10.5)	35 (27.1)	
	>50%	3 (15.8)	77 (59.7)	
Dural enhancement	AbsentPresent	07 (70.0)	31 (24.0)48 (52.7)	0.013
Edema	None	3 (15.8)	3 (2.3)	0.025
	<tumor volume	11 (57.9)	68 (52.7)	
	Equal to tumor volume	4 (21.1)	35 (27.1)	
	>tumor volume	1 (5.3)	23 (17.8)	
Cysts *	No	11 (57.9)	115 (89.1)	0.001
	Yes	6 (31.6)	14 (10.9)	
Subcortical involvement	Involved	18 (94.7)	94 (72.9)	0.044
	Not involved	1 (5.3)	35 (27.1)	
rCBV	Median (IQR)	1.8 [1.4–2.0]	2.6 [1.9–3.5]	0.001

IQR—interquartile range. * Two patients of IDH-mutant phenotype had hemorrhagic cysts.

**Table 3 jpm-13-00072-t003:** Multivariate analysis for IDH status.

Variable	Units	Odds Ratio	CI 95%	*p*-Value
Necrosis	None/<25%	Reference		
	>25%	0.04	[0.01; 0.17]	<0.001
rCBV	<=2.0	Reference		
	>2.0	0.12	[0.03; 0.56]	0.007

rCBV—relative cerebral blood volume.

## Data Availability

Not applicable.

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
