# Peer review of "Multiparametric Magnetic Resonance Imaging Correlates of Isocitrate Dehydrogenase Mutation in WHO high-Grade Astrocytomas"

_jpm, 2022, doi:10.3390/jpm13010072_

Round 1

Reviewer 1 Report

This work by Sahu et al., studies semantic Magnetic Resonance Imaging (MRI) features in gliomas comparing it with the gold standard biomarker IDH mutation status measured by immunohistochemistry. IDH mutation status is commonly used to categorise grade 4 gliomas (in mutant and wild-type ones) and to personalise treatments. The authors made a retrospective chart review of 148 grade-4 glioma patients (one of the largest cohorts studied).

Relative Cerebral Blood Volume (rCBV) and Necrosis accurately predicted the IDH mutation status. These findings were corroborated by the results of previous studies [references 16- 18]. Many MRI parameters did not significantly correlate with IDH status (Table S-2).

The study corroborates previous studies and, more important, documented new features related to IDH mutation status, such as enhancement pattern, dural enhancement, and subcortical involvement.

Then, the article provides relevant information but there are some issues that should be addressed:

- Improve the wording of the manuscript to show better the relevance of the new MRI features related to the IDH status.

- Information required on page 14 is completely absent (Author Contributions, Funding, Institutional Review Board Statement, Informed Consent Statement, Conflicts of Interest). This is unacceptable and it should be filled in before reconsideration.

Author Response

  1. Improve the wording of the manuscript to show better the relevance of the new MRI features related to the IDH status.

Thank you for your valuable comments. Improvement in the English language has been performed using Grammarly software.

  1. Information required on page 14 is completely absent (Author Contributions, Funding, Institutional Review Board Statement, Informed Consent Statement, Conflicts of Interest). This is unacceptable and it should be filled in before reconsideration.

Thank you for your valuable comments and for bringing this to our notice.

All the above necessary information is incorporated now in the corresponding section (page 14).

Author Contributions:

Conceptualization: Jayant S Goda, Arpita Sahu

Methodology: Nandakumar G Patnam, Sridhar Epari, Ayushi Sahay, Ronny Mathew, Amit Kumar Choudhari, Archya Dasgupta, Abhishek Chatterjee, Pallavi Pratishad and Prakash Shetty;

Validation: Jayant S Goda, Prakash Shetty and Tejpal Gupta

Formal analysis: Nandakumar G Patnam, Ronny Mathew, Abhishek Chatterjee and Pallavi Pratishad;

Investigation: Abhishek Chatterjee; Arpita Sahu

Data curation: Nandakumar G Patnam, Arpita Sahu, Sridhar Epari, Ayushi Sahay and Amit Kumar Choudhari;

Writing – original draft: Nandakumar G Patnam, Ayushi Sahay, Ronny Mathew, Archya Dasgupta and Abhishek Chatterjee;

Writing – review & editing: Jayant S Goda, Arpita Sahu, Prakash Shetty, Aliasgar Moiyadi and Tejpal Gupta; Visualization: Sridhar Epari, Amit Kumar Choudhari, Archya Dasgupta, Aliasgar Moiyadi and Tejpal Gupta;

Supervision: Arpita Sahu, Jayant S Goda, Sridhar Epari, Prakash Shetty and Aliasgar Moiyadi; Project administration: Arpita Sahu.

Funding: None

Institutional Review Board Statement: Consent waiver was obtained as it was a retrospective study.

Informed Consent Statement: Consent waiver obtained from IRB/ IEC.

Data Availability Statement: Not applicable

Acknowledgments: None

Conflicts of Interest: None

Reviewer 2 Report

GBM is the most aggressive human cancer that presents a poor prognosis. Each observation, including the one offered by imaging, is important in the manuscript. The authors clearly described their research and conclusions.
Please include the full name in the abstract - VASARI - Visually AcceSAble Rembrandt Images
Dots should be after citation, so, e.g. …..[1].
Under table 2 make the pace
Figure 6 . shift a description upper
p-vause should have "p" italicized,
authors did not include information from 336 lines, like author contributions, funding, etc. - please fil it.

Author Response

Thank you for your valuable comments. Please see our responses which are listed below.

  1. Please include the full name in the abstract - VASARI - Visually AcceSAble Rembrandt Images

We have included the full format in the abstract.

  1. Dots should be after the citation.

We have corrected all the dots and made them after the citation.

  1. Under table 2 make the space

We have inserted a space under table 2.

  1. Figure 6. shift a description upper

We have shifted the description up, for Figure 6.

  1. p-value should have "p" italicized

We have changed all p in p-value to italicized form.

  1. authors did not include information from 336 lines, like author contributions, funding, etc. - please fil it.

Thank you for bringing it to our notice. We have filled them all now.

Reviewer 3 Report

The authors propose their retrospective study on the MRI appearance correlation with IDH mutation in high grade gliomas.

The study involved 148 patients, 129 wild-types, and only 19 IDH-mutants.

They boldly concluded that the necrosis of <25%, and rCBV value of <2.0 suggested the IDH-mutant HGG.

The study design is poor, not just because of its retrospective nature, but also due to the uneven distribution, and possible bias.

Perfusion images were not included, and .

The manuscript also needs to be revised by the native English speaker, as some sentences appear rather confusing.

The study may lead the readership to the irrelevant conclusions and should be performed in the valid group of patients, before such recommendations should be given.

Author Response

Thank you for your valuable reviews and comments. Kindly look for our answers mentioned below.

  1. Perfusion images were not included

We have included the Dynamic susceptibility contrast-enhanced perfusion images and their post-processed images for calculating the rCBV values. We have mentioned this in our methodology too (please refer to page no. 3 lines 115- 118- under the MRI protocol section). Thank you.

  1. The manuscript also needs to be revised by a native English speaker, as some sentences appear rather confusing.

We really apologize for this issue. We have fully reviewed our paper with grammar correction software. Sorry for the inconvenience caused at first instance.

Round 2

Reviewer 3 Report

The authors resolved my concerns...